# Size-dependent strong metal-support interaction in TiO₂ supported Au nanocatalysts

Xiaorui Du[1,2,6], Yike Huang[1,3,6], Xiaoli Pan[1], Bing Han[1,3], Yang Su[1], Qike Jiang[4], Mingrun Li[2], Hailian Tang[1,5], Gao Li [2✉] & Botao Qiao [1,4✉]

The strong metal-support interaction (SMSI) has long been studied in heterogonous catalysis on account of its importance in stabilizing active metals and tuning catalytic performance. As a dynamic process taking place at the metal-support interface, the SMSI is closely related to the metal surface properties which are usually affected by the size of metal nanoparticles (NPs). In this work we report the discovery of a size effect on classical SMSI in Au/TiO₂ catalyst where larger Au particles are more prone to be encapsulated than smaller ones. A thermodynamic equilibrium model was established to describe this phenomenon. According to this finding, the catalytic performance of Au/TiO₂ catalyst with uneven size distribution can be improved by selectively encapsulating the large Au NPs in a hydrogenation reaction. This work not only brings in-depth understanding of the SMSI phenomenon and its formation mechanism, but also provides an alternative approach to refine catalyst performance.

[1] CAS Key Laboratory of Science and Technology on Applied Catalysis, Dalian Institute of Chemical Physics, Chinese Academy of Sciences, Dalian 116023, China. [2] State Key Laboratory of Catalysis, Dalian Institute of Chemical Physics, Chinese Academy of Sciences, Dalian 116023, China. [3] University of Chinese Academy of Sciences, Beijing 100049, China. [4] Dalian National Laboratory for Clean Energy, Dalian 116023, China. [5] College of Chemistry and Environmental Science, Hebei University, Baoding 071002, China. [6] These authors contributed equally: Xiaorui Du, Yike Huang. ✉email: gaoli@dicp.ac.cn; bqiao@dicp.ac.cn

Strong metal–support interaction (SMSI), coined by Tauster and co-workers in the late 1970s to describe the phenomenon that $TiO_2$ supported Pt-group metals (PGMs) will lose their capability to adsorb small molecules after high-temperature reduction[1,2], has been investigated extensively in the past decades. To date, it is widely recognized that SMSI is not only decisive in stabilizing active metals on supports[3–8], but also important in tuning catalytic performance and studying reaction mechanisms[9–12]. Unlike PGMs, of which the SMSI phenomenon on reducible oxides has long been acknowledged[13–21], Au had been regarded as an inert metal for SMSI in the past few decades due to its relatively lower work function and surface energy respect to PGMs[18,22,23].

However, a series of breakthrough in SMSI between Au nanoparticles (NPs) and different supports have been achieved in recent years. The $O_2$-induced SMSI between Au NPs and ZnO nanorods[24] and hydroxyapatite (HAP)[4] have been discovered successively, and an ultra-stable Au nanocatalyst was developed by tuning the SMSI in Au/HAP[5]. More recently, classical $H_2$-induced SMSI phenomenon in $Au/TiO_2$ has been uncovered which is highly conducive to improve either the catalyst stability[3] or even the activity[25]. These latest progresses in the SMSI between Au and supports not only highlighted the advantages of SMSI in improving the stability of Au nanocatalyst, but also stimulated motivation on seeking deeper understanding of SMSI which would arouse attention from both material and catalysis fields.

SMSI is a dynamic process taking place at the interface between Au NPs and the support, involving electron transfer and mass transport, both of which are closely related to the surface properties of Au NPs. Considering the prominent size-dependent catalytic performance of Au nanocatalysts[26–28], whether there is a size effect on SMSI remains an intriguing yet unaddressed issue. At nanometer scale, changes in metal particle size often lead to the variation in surface electronegativity, surface energy, and interfacial reactivity, thus affecting the SMSI process[18,22,29]. It has been proposed that the size of metal cluster may influence the encapsulation of PGMs by $TiO_2$[18,29,30], but so far no exclusive demonstration was reported. It was recently reported by Hutchings group that the smaller and poorly alloyed Pd-rich NPs are more susceptible to be encapsulated by $SnO_x$ than larger Pd–Sn alloy particles[31]. However, the universality of this trend was still uncertain due to the involvement of metal alloy particles.

Compared to Au NPs, supported PGM NPs are more prone to form SMSI: the SMSI of PGMs was not only realized upon high-temperature reduction but also observed in model catalysts in surface science studies where encapsulation of Pt, Pd, etc. can even occur upon ion sputtering or annealing in ultrahigh vacuum. However for supported Au NPs, no evidence of same phenomenon was found in surface science studies so far[22,32–34]. The less susceptibility of supported Au NPs for the occurrence of SMSI than PGMs together with their significant size effect in catalysis may provide an ideal model system to study the size effect in SMSI.

In this work, we report our discovery of the size-dependent SMSI in $Au/TiO_2$ catalyst system where SMSI is more prone to occur on large Au NPs (~9 and ~13 nm) than on small ones (~3 and ~7 nm). A surface tension dependent thermodynamic equilibrium model was established to explain this size effect. By utilizing this phenomenon, the hydrogenation reaction selectivity of $Au/TiO_2$ catalyst having an uneven particle size distribution was effectively improved. We believe the discovery in this work will find application in refining catalyst performance.

## Results

### Size-dependent SMSI in $Au/TiO_2$.
$Au/TiO_2$ catalysts with different Au particle size were synthesized by the colloidal deposition method[35–37]. Typically, colloidal Au NPs of ~3 and ~9 nm were synthesized by employing poly(vinylalcohol) (PVA) and oleylamine (OA) as protective agents, respectively, and depositing on commercial $TiO_2$ (P25, 58 $m^2 g^{-1}$) by an adsorption method. After removing ligands (Supplementary Fig. 1), the samples were further calcined at 450 and 400 °C to obtain Au/$TiO_2$ sample with other sizes. The size distributions were determined to be 3.4 ± 1.0, 7.0 ± 1.2, 9.4 ± 1.6, 13.6 ± 2.0 nm (Supplementary Fig. 2) and denoted as Au-3 nm, Au-7 nm, Au-9 nm, and Au-13 nm, respectively. Details of catalyst preparation are provided in method section and more sample information is presented in Supplementary Table 1.

To investigate their SMSI performances, samples were reduced at different temperatures, denoted as sample-HX (X represents the reduction temperature). Some of the reduced samples were re-oxidized under 10 vol% $O_2$/He for 1 h with a flow rate of 33.3 mL $min^{-1}$ at 400 °C and denoted as sample-HX-O400. Both Au particle size distributions (Supplementary Figs. 3 and 4) and crystal phase of $TiO_2$ (Supplementary Fig. 5) for each sample are barely changed by the high-temperature reduction or reoxidation treatment, providing an excellent testing ground for the following SMSI investigation.

Generally, classical SMSI between Au and $TiO_2$ is a dynamic phenomenon involving the suppression of small-molecule adsorption (such as $H_2$ and CO), electron transfer between the support and metal, and mass transport from the support to encapsulate metal NPs following high-temperature reduction, and a reversal of the preceding phenomena following re-oxidation. In situ diffuse reflectance infrared Fourier transform spectroscopy (DRIFTS) of CO adsorption on various catalysts following diverse heat treatments was first measured owing to its sensitivity to probe both adsorption property and electronic structure of the metal surface[17,24]. The results are shown in Fig. 1. Besides gaseous CO (2175 and 2120 $cm^{-1}$ bands), a strong adsorption at 2103–2105 $cm^{-1}$ were observed on all fresh samples, which can be attributed to CO adsorbed on metallic Au ($CO–Au^0$)[38]. The intensity of this $CO–Au^0$ band decreased gradually, along with the appearance of a band at ~2072 $cm^{-1}$ (ascribed to CO adsorption on negatively charged Au surface ($CO–Au^{x-}$))[38] whose intensity also decreased until completely disappeared with reduction temperature increasing. These indicate clearly the suppression of CO adsorption on, and the electron transfer to, Au surface, which are typical characters of the classical SMSI in $TiO_2$-supported metals[1–3]. Furthermore, after reoxidized at 400 °C for 1 h, CO adsorption was totally recovered on all samples; and the recovered adsorption bands were all slightly blue-shifted relative to that on corresponding fresh samples, evidencing the reversible surface adsorption capacity as well as the reversible electron transport between Au and support following reoxidation treatment.

The CO-DRIFT results demonstrate that on all samples with different Au size SMSI occurred after $H_2$ reduction[3]. However, a notable distinction is that the reduction temperatures at which the CO adsorption completely disappeared are totally different. For the Au-3 nm sample (Fig. 1a), the CO adsorption disappeared completely only after reduction at 600 °C while for Au-7 nm sample it disappeared at reduction at 500 °C (Fig. 1b). For samples with much larger Au NPs (Au-9 nm and Au-13 nm), CO adsorption totally disappeared at a much lower reduction temperature of 400 °C (Fig. 1c, d). The normalized CO coverage with different reduction temperatures on various samples was presented in Supplementary Fig. 6, where the size-dependent CO coverage on Au NPs is clearly identified. The distinct difference in CO-DRIFT results may suggest a prominent size effect on SMSI between Au and $TiO_2$ which has not been disclosed so far. The size effect of mass transport from $TiO_2$ to encapsulate Au NPs

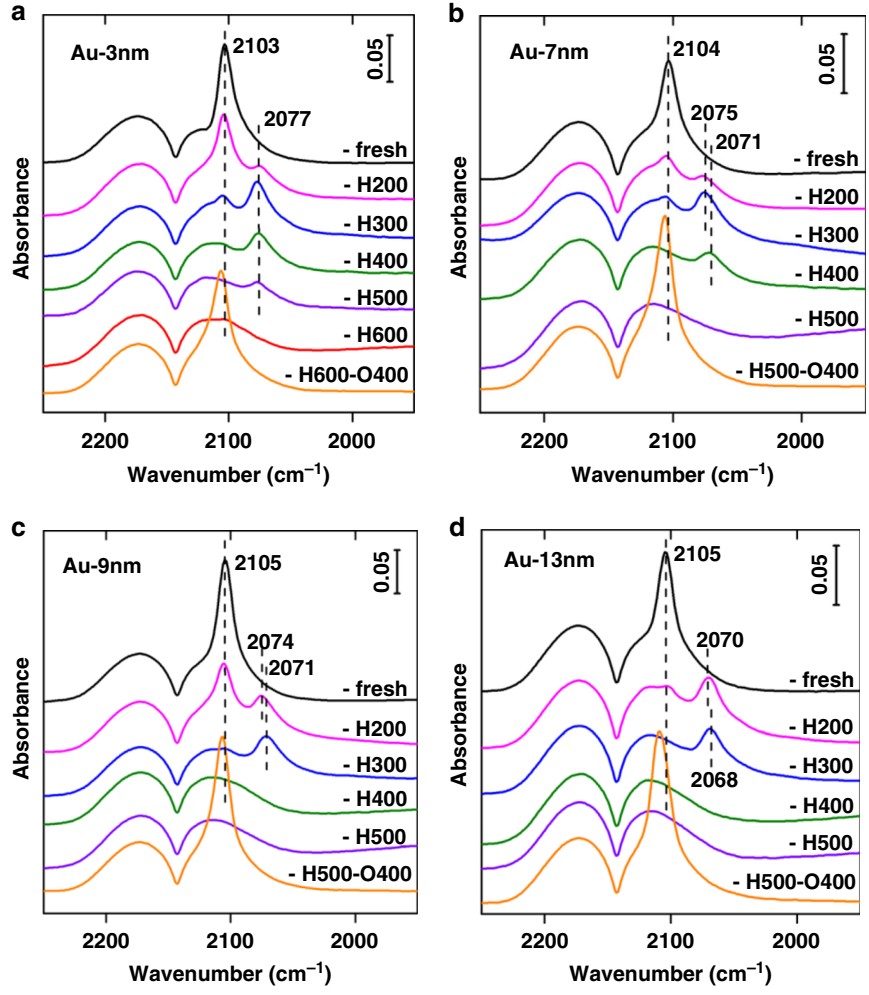

**Fig. 1 In situ DRIFT spectra of CO adsorption over Au/TiO$_2$ with different particle size. a** Au-3 nm, **b** Au-7 nm, **c** Au-9 nm, and **d** Au-13 nm. The notation of -fresh represents the as-synthesized samples, the notation of -HX represent samples reduced by 10 vol% H$_2$/He under different temperatures, and the notation of -HX-O400 represent samples reoxidized at 400 °C after reduction.

following reduction treatment was also confirmed by high-resolution transmission electron microscopy (HRTEM). The bear Au NPs with different size can be clearly observed for the as-synthesized samples, as shown in Fig. 2a, c, e, g. After reduction at the temperatures where the CO adsorption completely disappeared, the Au NPs were completely encapsulated by a thin layer in all samples, as shown in Fig. 2b, d, f, h. Electron energy loss spectroscopy (EELS) examinations further revealed the low-valent Ti in the encapsulation layers on Au-3 nm-H600 and Au-9 nm-H400 samples[3], Supplementary Fig. 7, evidencing the coverage by TiO$_{2-x}$ species, similar to that of the Pd/TiO$_2$ system[39]. Of more importance, the Au NPs in Au-3 nm sample reduced at 400 °C were only partly encapsulated, which was clearly confirmed by EELS mapping analysis for Au-3 nm-H400 (Fig. 3). As can be seen, both the mapping result (Fig. 3c) and the extracted EELS spectra (Fig. 3e) of Ti on Au NPs suggest that the Au NPs on Au-3 nm-H400 were not completely encapsulated (for example in region 4 the Ti signal is obviously weaker than that in 2 and 3, and more importantly, in region 5 there's no detectable Ti signal). All these results unambiguously demonstrate the presence of size effect in SMSI.

In the case of the re-oxidized samples where CO chemisorption was restored, the thin overlayers have retreated (Supplementary Fig. 8). In addition, with Au particle size increase, a slight red shift of the CO–Au$^{x-}$ band can be observed for the samples

reduced at a same temperature. For instance, after reduced at 300 °C, the CO–Au$^{x-}$ bands on Au-3 nm-H300, Au-7 nm-H300, Au-9 nm-H300, and Au-13 nm-H300 are at 2077, 2075, 2071, and 2068 cm$^{-1}$, respectively (Fig. 1 and Supplementary Fig. 9). This result indicates an increase of electron transfer with Au size, which was also verified by electron paramagnetic resonance (EPR) measurements. As shown in Supplementary Fig. 10, the differences in both intensity and width of the $g = 1.94$ signal (ascribed to surface Ti$^{3+}$ species after high-temperature reduction[40–42], see Supplementary Information for more details) between two samples suggest that Au-9 nm-H400 has more surface Ti$^{3+}$ species than that of Au-3 nm-H400, further demonstrating the size effect on SMSI. It should be noted that the electron transfer under SMSI condition is distinctly different from the electron transfer under electronic metal-support interaction (EMSI)[43,44] condition, where the largest charge transfer was observed on metal particles of about 1.5 nm and/or smaller than 4 nm[45–47]. The EMSI naturally forms in some special catalyst systems without any treatments while the SMSI only occurs upon special treatments, such as high-temperature reduction (the classical SMSI), high-temperature oxidation[4,8,24], or during reaction[12,48]. In this work we studied the classical SMSI where high-temperature reduction by H$_2$ is needed, thus the electron transfer between the support and Au is much easier than that in the case of EMSI.

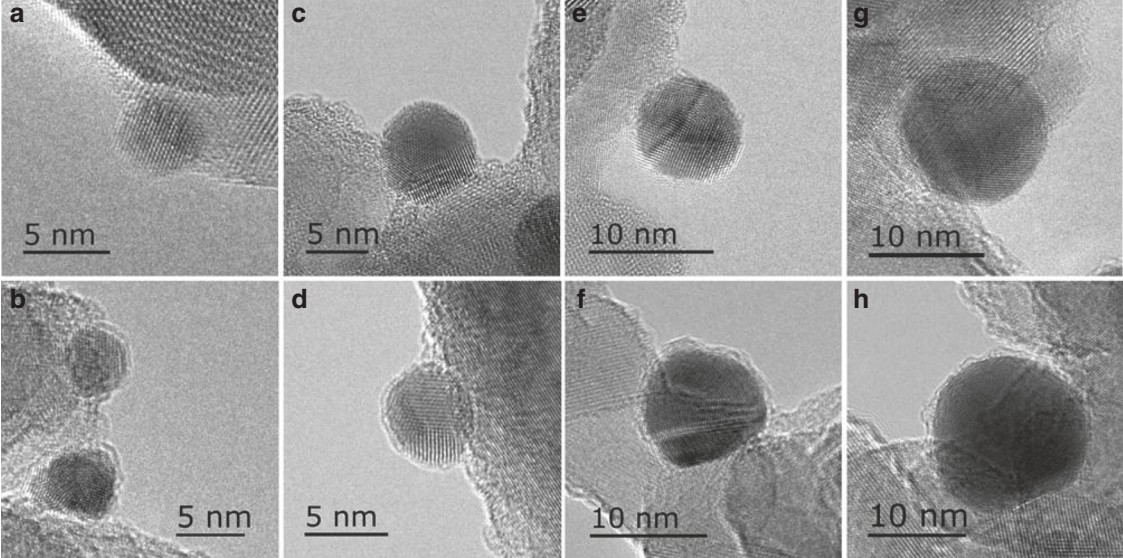

**Fig. 2 Representative HRTEM images of different samples. a** Au-3 nm, **b** Au-3 nm-H600, **c** Au-7 nm, **d** Au-7 nm-H500, **e** Au-9 nm, **f** Au-9 nm-H400, **g** Au-13 nm, and **h** Au-13 nm-H400.

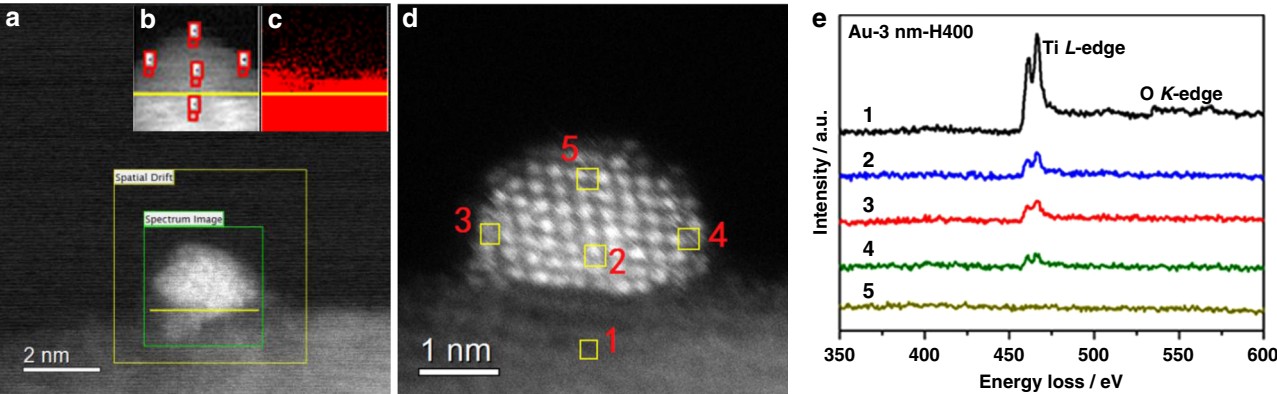

**Fig. 3 EELS mapping analysis of Au-3 nm-H400 sample. a** Survey image for the EELS mapping test, **b** the EELS spectrum image, **c** the EELS elemental map for Ti, **d** the corresponding high-angle annular dark-field scanning transmission electron microscopy (HAADF-STEM) image, and **e** the extracted EELS spectra (background subtracted) of the selected positions (red mark) in (**b**), as also marked by yellow squares in (**d**). The yellow lines in (**a–c**) mark the interface of Au NP and $TiO_2$.

Above studies unambiguously revealed the existence of size effect in the SMSI on $Au/TiO_2$ catalysts in terms of not only electron transfer but also mass transport between Au NPs and $TiO_2$. One may question whether the geometry of the NPs has influence on the SMSI, which we do not have an answer so far. An independent work to systemically study the effect of geometric structure is needed which is beyond the scope of this work. However, we would note that in this work, the effect comes mainly from size rather than geometry because at least for ~3 and ~7 nm Au NPs the structure is same (truncated octahedron with Au {111} and {100} facets mainly exposed, Supplementary Fig. 11).

**Mechanism of the size-dependent SMSI in $Au/TiO_2$.** The encapsulation occurred during SMSI can be considered as a wetting process of the Au NPs by reduced titanium oxide ($TiO_{2-x}$). As reported by Fu et al.[18], the encapsulation process is mainly determined by the surface tensions. According to surface tension balance Eq. (1), a larger surface tension of the metal than that of $TiO_2$ support is expected. This is often the case for $TiO_2$ supported PGMs (Supplementary Table 2). But for Au, a

relatively low surface tension ($\gamma_{Au} = 1.51\,J\,m^{-2}$) was traditionally regarded as the main reason for not forming SMSI[18,23]. However, recent discovery of SMSI in $TiO_2$ supported Au[3], or Ag with an even lower surface energy than Au[3,49], implies either the previously reported $TiO_2$ surface tension (1.3–1.9 $J\,m^{-2}$, Supplementary Table 2) is overestimated or other detailed structure factors may need to be focused on or be reconsidered. The experimentally measured relatively low surface energy of $TiO_2$ (0.5–1.7 or <0.7 $J\,m^{-2}$, Supplementary Table 2) suggests the former might be more probable. Therefore, the case Au possessing higher surface tension than $TiO_2$ being wetted by $TiO_2$ is possible where the minimization of surface free energy of Au NPs is the major driven force. On the other hand, in nanoscale, size-dependence of surface tension is significant and has been extensively studied since Tolman[50]. Although both surface tension correlated with particle size positively and negatively were reported, at high temperatures a positive correlation between particle size and surface tension is more plausible as the contribution of surface entropy is non-negligible, (Supplementary Fig. 12, see Supplementary Information for more details)[51]. Therefore for an estimation of driving force, surface tension increases with NP size, Supplementary Eq. (1). In other words,

larger Au NPs with higher surface energy has stronger tendency to be wetted by TiO$_2$.

$$\gamma_{Au} = \gamma_{int} + \gamma_{TiO_{2-x}}. \tag{1}$$

Assuming the encapsulation reaction mainly occurred at the interface of Au/TiO$_2$, the dynamic reaction process is further explored by establishing a surface tension dependent thermo-dynamic equilibrium model, as shown schematically in Fig. 4b, which is directly extracted from Fig. 4a. Surface tension balance (Eq. (1)) can always be satisfied at contact point. We assume that in the scale of NP size of interest, overlayer species TiO$_{2-x}$ are abundant, therefore at any thermodynamically equilibrated states the encapsulation degree $\theta$ is only the function of surface tensions $\gamma_{Au}$, $\gamma_{int}$, $\gamma_{TiO_{2-x}}$, and $r$, the radius of NP. Start from Eq. (1), after formulation (see Supplementary Information for detailed process), the relationship between $r$ and $\theta$ is obtained, as expressed in Supplementary Eq. (8). The positive correlation between $r$ and $\theta$ discovered in experiments is successfully reproduced by solving Supplementary Eq. (8) numerically (solutions are plotted in Supplementary Fig. 13). Therefore, the established thermo-dynamic dynamic equilibrium model above provides a concrete interpretation of the experimental evidenced size effect in SMSI of Au/TiO$_2$.

**Catalytic application of the size-dependent SMSI.** It is known that in many reactions the catalytic performance of supported gold catalysts is size dependent[26–28,52,53]. The usually uneven particle size distribution may give rise to different products thus

lowering the selectivity. The finding of size effect on the SMSI of Au/TiO$_2$ may hence provide an approach to finely tune the catalytic performance by selectively encapsulating larger Au NPs. To verify this, we synthesized a model catalyst (Au-3 + 9 nm) by depositing both 3 and 9 nm Au NPs onto TiO$_2$, as shown in Supplementary Fig. 14, and studied their catalytic performance by using the chemoselective hydrogenation of 3-nitrostyrene as probe reaction. This is a structure sensitive reaction in terms of selectivity due to the different adsorption models of the 3-nitrostyrene on different size of metals, as suggested in Supplementary Fig. 15[54]. On the other hand, whether the activity is size sensitive or not remains unclear because the rate determination step could be the H$_2$ activation which, however, might be different on different size of metal surfaces. Accordingly, in this work we only focus on the size-dependent selectivity. As shown in Table 1, Au-3 nm has a relatively high selectivity of ~78% (entry 1) while Au-9 nm yields much lower selectivity (~39%, entry 2) with similar conversion, suggesting that larger Au NPs are less selective. This is in agreement with the structure sensitivity of 3-nitrostyrene hydrogenation. The Au-3 + 9 nm sample shows a medial selectivity between those of Au-3 and Au-9 nm (54.7% in Entry 3). However, after reduced at 400 °C, the selectivity was improved significantly (96.1% in Entry 4), which is same to that of Au-3 nm-H400 (95.5% in Entry 5). The selectivity increase in Au-3 + 9 nm-H400 can be explained rationally by the size-dependent SMSI. On the one hand, the large Au particle was almost completely encapsulated, as confirmed by the significantly decreased activity of Au-9 nm-H400 (Entry 6 in Table 1) and the HRTEM images, Supplementary Fig. 16. On the other hand, the slightly increased selectivity in Au-3 + 9 nm-H400 and Au-3-H400 compared with Au-3 nm must have originated from an electronic effect upon reduction due to the favorable adsorption of -NO$_2$ group on electron-rich sites[54–57]. This is another advantage of utilizing the SMSI to produce negatively charged active sites.

The normalized activity of Au-3 + 9 nm-H400 was also measured (Supplementary Table 3), and the calculated TOF based on the total amount of Au is 11.9 h$^{-1}$ at 110 °C, which is higher than that of Au-3 nm-H400 (7.1 h$^{-1}$) and Au-9 nm-H400 (1.3 h$^{-1}$). The catalyst stability of Au-3 + 9 nm-H400 was studied as well. It shows that the sample is quite stable as in five cycles of reaction the activity decrease is minor and selective change is undetectable, Supplementary Fig. 17. The used sample was characterized by HAADF-STEM and HRTEM. The comparison of HAADF-STEM and HRTEM images of the Au-3 + 9 nm-H400 sample before (Supplementary Fig. 16) and after (Supplementary Fig. 18) reaction shows that the increase of Au particle size was hardly observed. Meanwhile, the situation where large Au particles were encapsulated while the small particles were only

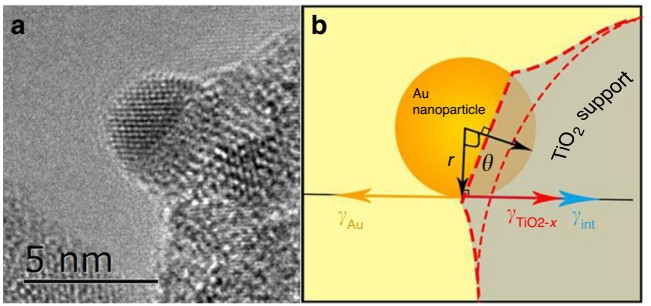

**Fig. 4 Establishment of the thermodynamic equilibrium model.**
**a** Representative HRTEM image of Au-3 nm-H400. **b** Schematic of encapsulation of Au particle (in a spherical segment) by TiO$_2$ support. $r$ (black arrow) represents the radius of Au particle, and $\gamma_{Au}$ (orange arrow), $\gamma_{TiO_{2-x}}$ (red arrow), and $\gamma_{int}$ (blue arrow) are the surface tensions of the Au particle, TiO$_2$ support, and the interface between them. The thick and thin dashed red line in (**b**) indicate the outline of the encapsulation layer and the outline of the original TiO$_2$ support, respectively.

**Table 1 Chemoselective hydrogenation of 3-nitrostyrene using different catalysts.**

| Entry | Catalyst | Conversion (%) | Selectivity for 3-vinylaniline (%) | Selectivity for 3-ethylnitrobenzene (%) | Selectivity for 3-ethylaniline (%) |
|---|---|---|---|---|---|
| 1 | Au-3 nm[a] | 15.4 | 78.3 | 16.0 | 5.7 |
| 2 | Au-9 nm[b] | 14.9 | 38.9 | 51.4 | 9.7 |
| 3 | Au-3 + 9 nm[c] | 24.8 | 54.7 | 45.3 | 0 |
| 4 | Au-3+9nm-H400[c] | 28.3 | 96.1 | 3.9 | 0 |
| 5 | Au-3nm-H400[a] | 12.0 | 95.5 | 4.5 | 0 |
| 6 | Au-9nm-H400[b] | 3.5 | 100 | 0 | 0 |

Reaction conditions: $T$ = 110 °C, $P_{H_2}$ = 1.0 MPa, reacted for 4 h.
[a] 40 mg of catalyst (Au loading of 0.51%).
[b] 20 mg of catalyst (Au loading of 1.45%).
[c] 20 mg of catalyst (total Au loading of 1.46%).

partially covered was kept after the reaction. The above catalytic results clearly demonstrate the potential applications of size-dependent SMSI in catalysis area.

## Discussion

We have discovered an unanticipated size effect of SMSI on Au/TiO$_2$, and successfully explained it by establishing a thermo-dynamic equilibrium model where the positive correlation of the degree of encapsulation of Au NPs with their size was obtained. Furthermore, the size effect in SMSI can be utilized to tune catalytic selectivity of Au/TiO$_2$ for hydrogenation of 3-nitrostyrene by selectively encapsulating the large Au NPs as well as tuning the electronic property of the small ones. This work may bring an in-depth understanding of SMSI mechanism, and provide an alter-native approach to refine catalytic performance by tuning the SMSI state.

## Methods

**Materials**. All the chemicals were commercially available and used without further purification. Hydrogen terachloroaurate (IV) hydrate (HAuCl$_4$·4H$_2$O) was purchased from Tianjin Fengchuan Chemical Reagent Technologies Co., Ltd. sodium borohydride (NaBH$_4$, 98%) were purchased from Alfa Aesar. P25 (TiO$_2$) was purchased from Evonik Degussa, with the purity of 99.5% and the specific surface area of 58 m$^2$ g$^{-1}$, and used without further treatment. PVA, molecular weight ($M_w$) 10,000, 80% hydrolyzed] was purchased from Aldrich. OA (80-90%) and 3-nitrostyrene was purchased from Aladdin Company. Toluene and o-xylene were purchased from Sinopharm Chemical Reagent Co., Ltd. Ethylene glycol (AR, 99.8%) and cyclohexane (AR, 99.7%) were purchased from Damao Chemical Reagent Factory. Ethanol (AR, 99.7%) was purchased from Tianjin Fuyu Fine Chemical Co., Ltd.

**Synthesis of Au-3 nm and Au-7 nm samples**. Typically, 6.67 mL of PVA solution (0.50 mg mL$^{-1}$) was added to a 50 mL aqueous HAuCl$_4$ solution (0.0254 mmol of Au, Au/PVA = 1.5:1 mg mg$^{-1}$) at room temperature under vigorous stirring. After 10 min, 1.27 mL of 0.10 mol L$^{-1}$ NaBH$_4$ solution was rapidly injected into the solution which turn to dark orange-brown immediately, indicating the formation of gold colloid. A 1.00 g of support P25 was then added and the mixture was continuously stirred for 6 h. The solids were collected by filtration, washed with deionized water, dried at 60 °C overnight, and then calcined at 300 °C for 1 h under air flow (55 mL min$^{-1}$) in air using a heating ramp of 5 °C min$^{-1}$. The obtained sample was denoted as Au-3 nm. The Au content was determined to be 0.59 wt% by inductively coupled plasma optical emission spectrometer (ICP-OES).The sample with average Au particle size of 7 nm (Au-7 nm) was prepared by the same method with that of Au-3 nm. While for Au-7 nm, the loading amount of Au was doubled and the calcination condition was switched to 300 °C for 1 h under air flow (55 mL min$^{-1}$) with heating ramp of 10 °C min$^{-1}$ and further 450 °C for 2 h with heating ramp of 5 °C min$^{-1}$ under air flow (55 mL min$^{-1}$) in air. The Au content was determined to be 1.20 wt% for Au-7 nm.

**Synthesis of Au-9 nm and Au-13 nm samples**. For the Au/TiO$_2$ catalyst with average Au particle size of 9.4 nm (Au-9 nm), OA was used as protecting agent[36]. Typically, HAuCl$_4$ (0.0254 mmol of Au) and 10 mL OA were mixed into a glass pressure vessel (Synthware). The sealed vessel was evacuated for 15 min, and the mixture was turned to a clear transparent yellow solution. This solution was subsequently heated to 150 °C and kept for 2 h under stirring before it was cooled to room temperature. The obtained OA capped Au NPs in the solution were precipitated using ethanol, separated via centrifugation, further purified using a cyclohexane–ethanol solution, and finally dissolved in 30 mL of cyclohexane to forming a transparent wine red colloidal gold solution. Totally, 350 mg of support P25 was dispersed in another cyclohexane solution by sonication, and then the Au NPs dispersion was added. The mixture was left stirring for 6 h. The solid was recovered by evaporating hexane naturally, dried at 60 °C overnight, and calcined at 350 °C for 1 h under air flow (55 mL min$^{-1}$) using a heating ramp of 2 °C min$^{-1}$. The Au content was determined to be 1.20 wt% by ICP-OES. The sample with average Au particle size of 13.6 nm (Au-13 nm) was prepared by the same method with that of Au-9nm. While for Au-13nm, calcination condition was switched to 400 °C for 1 h under air flow (55 mL min$^{-1}$) in air using a heating ramp of 10 °C min$^{-1}$. And the Au content was determined to be 1.49 wt% by ICP-OES.

**Synthesis of Au-3 + 9 nm sample**. The Au/TiO$_2$ catalyst with a mixed particle size distribution was synthesized by combining the synthetic route of Au-3 nm and Au-9 nm, and the loading amount of Au was adjusted as needed. The Au/TiO$_2$ sample with average Au particle size of about 3 nm was first prepared. After dried at 60 °C for 24 h, the obtained product was dispersed in a cyclohexane solution by sonication. Then an OA capped Au NPs dispersion in cyclohexane was added

under vigorous stirring. The solid was recovered by evaporating hexane, dried at 60 °C overnight, and calcined at 350 °C for 1 h under air flow (55 mL min$^{-1}$) using a heating ramp of 2 °C min$^{-1}$. The final product was named as Au-3 + 9 nm, and its Au content was determined to be 1.46 wt% by ICP-OES, in which the content of the particle size of about 3 nm was 0.25 wt%.

**Pretreatment of Au/TiO$_2$ samples**. The as-synthesized Au/TiO$_2$ samples were heat treated to investigate their SMSI performances. Generally, the samples were reduced under 10 vol% H$_2$/He for 1 h at a flow rate of 33.3 mL min$^{-1}$ at different temperature either in situ in the DRIFT instrument or in a specialized apparatus for reduction. The reduced samples were denoted as sample-HX where X was the reduction temperature. To investigate the reversible of the SMSI of Au/TiO$_2$, some of the reduced samples were further oxidized under 10 vol% O$_2$/He for 1 h at a flow rate of 33.3 mL min$^{-1}$ at 400 °C, which were denoted as sample-HX-O400.

**Characterization**. The loading levels of Au were determined by ICP-OES on a PerkinElmer Optima 7300 DV instrument. The Brunauer–Emmett–Teller specific surface area of P25 was measured by N$_2$ nitrogen adsorption-desorption isotherms at 77 K using a Micromeritics ASAP-2460 analyzer. Before measurements, the sample was degassed in vacuum at 300 °C for 5 h. Thermogravimetric analysis was conducted on a SETARAM SETSYS 16/18 thermal analyzer. The samples were heated to 800 °C in air flow (100 mL min$^{-1}$) with a ramp rate of 10 °C min$^{-1}$. FT-IR spectrum were recorded on a BRUKE EQUINOX55 Fourier transform infrared spectrometer under transmission mode. All samples were ground and mixed with KBr and then pressed to form pellets. The background spectrum was subtracted from the sample spectrum. Powder X-ray diffraction was performed with a PNAnaly6tical X'Pert PRO diffractometer with monochromatized Cu-Kα radiation source (λ = 0.15432 nm). DRIFTS were collected by using a BRUKER Vertex 70 spectrometer equipped with an MCT detector. The spectrum was obtained by collecting 32 scans at a resolution of 4 cm$^{-1}$ at temperature of 25 °C. A certain amount of a powder sample was loaded into the cell with a ZnSe window which can work at high temperature. For each CO adsorption round of fresh catalysts, the sample was purged with He for 0.5 h at 120 °C, and the background spectrum was recorded after cooled to 25 °C. Then a mixture gas of 3 vol% CO/He was intro-duced into the reaction cell with a rate of 33.3 mL min$^{-1}$, and the CO-DRIFT was collected until the sample was saturated adsorbed by CO. For the CO adsorption round of samples need pretreatment, the sample was in situ reduced under 10 vol% H$_2$/He (or oxidized under 10 vol% O$_2$/He) flow with a rate of 33.3 mL min$^{-1}$ for 1 h at different temperature (denoted as -HX or -OX in sample name, respectively). After purged with He and cooled to 25 °C, the background spectrum was recorded and subsequently the mixture gas of 3 vol% CO/He was introduced for collecting CO absorption spectrum. HRTEM, and HAADF-STEM images were recorded using a JEOL JEM-2100F microscope operated at 200 kV. The EPR spectra were recorded at 110 K with a Bruker A200 spectrometer. The EELS analysis for Au-3nm-H600 and Au-9 nm-H400 was conducted on an Grand ARM300F microscope equipped with a Quantum ER965 type EELS accessory. The chemical compositions of the covering layer of Au NPs were characterized by directly putting electron beam at the Au NPs in a STEM mode. The EELS mapping analysis for Au-3 nm-H400 was performed on a JEOL-ARM200F scanning trans-mission electron microscope with a probe spherical aberration (Cs) corrector working at 200 kV. The instrument was equipped with a Gatan Quantum 965 image filter system.

**Catalytic performance measurement**. Catalytic performance of the selective hydrogenation of 3-nitrostyrene was carried out in a stainless steel autoclave equipped with a pressure gauge and magnetic stirring. Before reaction, a mixture of 3-nitrostyrene (0.12 M), toluene and o-xylene (0.06 M), totaled 3 mL was put into the vessel. Certain amounts of catalysts were then introduced into the autoclave. After being sealed, the autoclave was flushed with hydrogen for five times and then pressurized at 1.0 MPa. The reactor was heated to a certain temperature in an oil bath. After reaction, the product was analyzed by gas chromatography/mass spectrometry.

## Data availability

The data that support the findings of this study are available within the paper and its Supplementary Information, and all data are available from the authors on reasonable request.

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

## Acknowledgements

This work was supported by National Key Projects for Fundamental Research and Development of China (2016YFA0202801), Strategic Priority Research Program of the Chinese Academy of Sciences (XDB17020100), National Natural Science Foundation of China (21972135 and 21776270), DNL Cooperation Fund, CAS (DNL180403), LiaoNing Revitalization Talents Program (XLYC1807068), Innovation Fund of DICP (DCLS201704), and Natural Science Foundation of China and Japan Society for the Promotion of Science Cooperative Research Project (21961142006). X.D. thanks the support by China Postdoctoral Science Foundation (2018M641725). Q.J. acknowledges support from the DNL Cooperation Fund, CAS (DNL201903) and the National Natural Science Foundation of China (No. 51701201).

## Author contributions

X.D. performed the synthesis, most of the material characterizations, and the catalysis tests. Y.H. established the thermodynamic equilibrium model and carried out the

formulation. X.P. and Y.S. carried out the STEM and HRTEM characterizations. B.H. and H.T. participated in some experiments. Q.J. and M.L. carried out the EELS test. X.D. and B.Q. wrote the paper. G.L. and B.Q. supported and supervised the project.

## Competing interests

The authors declare no competing interests.
