## [Peer Review File · Nature Communications]

REVIEWER COMMENTS

Reviewer #1 (Remarks to the Author):

In this work, the authors present the study on the influence of particle size on metal-support interaction in Au/TiO₂ catalysts. It is claimed that larger Au nanoparticles tend to be encapsulated by TiO₂ overlayers during high-temperature reduction treatment. However, the discussion on the metal-support interaction on the catalytic behavior of various Au/TiO₂ catalysts is not convincing. The control experiments were not well performed. Therefore, I do not think this current manuscript meets the requirements of nature communications and more works are required to reinforce their results and conclusions.

1) Au nanoparticles were prepared by two methods, using different ligands. The calcination temperature is 300 °C in air, which may not be enough to completely remove the organic ligands covered on the surface of Au nanoparticles. The authors should make sure the surface of the Au nanoparticles is clean. Following this comment, the calcination temperature for Au-7nm sample is different from the Au-3nm. It is well known that the geometric structure of metal nanoparticles can evolve during thermal treatment and is related to the gas environment. Indeed, the particle shape of Au nanoparticles of various sizes is different, as indicated by TEM images. So, how the geometric structure may influence the SMSI?

2) In the formulation of the thermodynamic equilibrium for the encapsulation of Au NPs, it is assumed that the Au nanoparticle is a spherical particle. However, they are not, as can be seen in the TEM images. The authors should try to establish a more realistic model and should take into account the Au-TiO₂ interface which is also related to the size of Au NPs.

3) It is claimed that Au-3nm sample reduced at 400 °C by H₂ is only partially encapsulated by the support. However, the TEM is not sufficient to provide such a conclusion since the contrast of the TiO_x overlayers are not very clear in TEM images. The authors should carry out EELS mapping to show the coverage of Au nanoparticles, as complementary results to the CO-IR.

4) The hydrogenation of C=C bonds is not a structure sensitive reaction while the hydrogenation of -NO₂ is a structure sensitive reaction. In Ref.46, it is shown that, a selective catalyst for hydrogenation of -NO₂ to -NH₂ is more favorable with metal catalysts with a relatively small exposed surface. In some works from Somorjai's group, there are also some papers on the discussion of the structure-sensitivity of hydrogenation of olefins.

5) In Table 1, the author shows the reactivity of Au-3+9nm-H400 and Au-3nm-H400. How about the Au-9nm-400H? The higher activity of Au-3+9nm-H400 could be caused by the reduction treatment since the Au-3nm and Au-9nm samples were prepared by calcination in air. Besides, it seems there is a synergistic effect between Au NPs of 3 nm and 9 nm since the Au-3+9nm-H400 sample shows much higher activity than the Au-3nm and Au-9nm. However, the Au-3nm-H400

sample shows lower activity than the Au-3nm, which could be related to the partial encapsulation of Au nanoparticles, according to Ref.46. In my opinion, the SMSI effect is not sufficient to explain such results. More detailed studies should be carried out to verify such effects and give a more sound discussion.

Reviewer #2 (Remarks to the Author):

In this work the influence of Au NP size on the degree of H₂ induced SMSI encapsulation by a TiO₂ support was examined. This is an area of catalysis that has seen interest since the 1970's, and recently has seen significant interest. The SMSI encapsulation state on Au nanoparticles was found to form at a higher temperature for 3 nm Au NPs compared to 9nm and 13 nm AuNPs (600oC vs. 400oC in H₂), suggesting that larger Au NPs are more prone to being encapsulated within the TiO_x layer. On the basis of previously reported size dependent changes to the Au surface energy, the authors developed a thermodynamic model that supports a positive correlation between the degree of the SMSI encapsulation and the Au NPs size. The influence of the size-dependent SMSI formation on the chemoselectivity in hydrogenation of 3-nitrostyrene was also explored.

Overall this paper has interesting measurements and findings. The Au particle size dependent phenomenon may not be completely surprising, but if well analyzed the paper could be published in Nature Comm. However, there are issues throughout the analysis that should be thoroughly addressed prior to publication.

1. It would be useful for the authors to integrate the area of the CO stretch for adsorbed species on Au vs reduction T to show the size dependent decrease in CO coverage clearly. This would be better achieved if the gas phase CO background were subtracted. Also, the units on the IR data should be checked (as shown this is not A.U., this has a specific meaning (Anal. Chem. 2007, 79, 10, 3912-3918)).
2. Correlation between the CO band position and EPR signature is used as evidence of the size effect. First, there is a higher weight loading of Au on the larger particle sample – so this is not a fair comparison. Second, the charge transferred to Au will likely delocalize throughout the particle (it is a metal). To have a more “anionic” Au particle of 9 nm diameter compared to 3 nm diameter, 27x more electrons would need to be transferred due to the particle volumes. Given then 2x higher weight loading in the 9 diam case, more than 50x electrons transferred is needed to make 9 nm Au more anionic than 3 nm Au. This would likely cause bulk reduction of TiO₂? This argument relating EPR to CO IR needs work (see Nature Materials volume 15, pages284–288(2016)).
3. I was surprised not to see any discussion of recent work from Pd/TiO₂ that suggests a specific structure of TiO_x overlayers on these metals and shows their thermodynamic stability (Nano Lett. 2016, 16, 7, 4528-4534). Is the overlayer structure expected to be the same for Au? This should be critical for defining the free energy of the encapsulated state. Also, the model presented is for a rounded Au surface. In actuality is it the higher density of steps and edges for smaller sizes that causes the apparent Au size dependent SMSI behavior?
4. The reactivity measurements section needs additional discussion. Most importantly, there is a

discrepancy between the catalytic conversions measured for Au-3nm and Au-9 nm to that of the Au-3nm+9nm mixture after treatment at 400°C, which could be clarified. The catalytic conversions are almost the same for the 40 mg Au-3nm with 0.51% Au and 20 mg Au-9nm with 1.45% Au (entry 1 and 2, Table 1). While the active surface area measurements are missing, it seems that based on the total Au loading the active site for Au-3 is 1.42 that of the Au-9 nm (a reasonable assumption). However, for the Au-3nm+Au-9nm mixture consisted of 0.25% Au-3 nm (and 1.21% Au-9 nm based on the total 1.46% Au) after the formation of the SMSI state following reduction at 400°C, which substantially covers the active sites on the 1.21% Au-9nm, why did the conversion increase to 28.3%? It would be useful to see the reactivity of the Au-3nm+Au-9nm mixture sample without 400 C pre-treatment to show it was a simple additive mixture of the behavior of the individual samples. Also, showing the reactivity of the Au-9nm sample after 400 C reduction would be useful. This and added discussions based on exposed surfaces area of each particle size would be very helpful to clarify the influence of size dependent SMSI on reactivity.

Reviewer #3 (Remarks to the Author):

Au/TiO₂ catalysts with different Au particle sizes have been studied for the SMSI effect. This effect involves (partial) coverage of the metal surface with TiO_x suboxides to affect the number and performance of active gold metal sites. It is reported that the larger the Au particle size the stronger the SMSI effect, i.e. the lower the reduction temperature at which the SMSI is substantial. This finding in itself is very surprising and if substantiated is sufficient to warrant publication in Nature Communications. TEM, EELS, EPR, CO-FTIR seem convincing in establishing the extent and nature of the SMSI as function of Au-NP size and of the reduction temperature. However, some concerns have to be addressed before publication.

1. As mentioned, it is very surprising that SMSI would be more pronounced for larger Au-NP. This finding is in contrast with conclusion from most recent review – Nat. Catal. 2 (2019) 955–970 where it is stated that for particles smaller than 4 nm SMSI effects are larger. The authors might want to include this in their introduction.
2. For rationalizing that larger Au-NP are more prone to SMSI it is stated that larger Au-NP have higher surface energies, see Suppl Fig. 9. This statement is contrary to many papers from which I mention only two: Bulletin of the Russian Academy of Sciences: Physics volume 80, pages 698–701 (2016) and Nanomaterials 2020, 10, 484; doi:10.3390/nano10030484, the latter paper about Au specifically. Smaller particles inevitably have surface atoms with lower coordination numbers at their surfaces which coincides with higher solid-gas surface tensions and not lower ones. So the authors have a serious problem at hand to prove that all other studies except for quoted refs 42-44 that are from a single research group and do not provide enough evidence in my opinion.
3. My second serious concern is that the catalysis studies are not sufficient to draw conclusions. In fact these studies are of a preliminary nature. First, it is essential to report conversion/selectivity versus time curves by variation of the time that the experiment lasts. From that the authors can draw conversion –

time curves and via tangent method calculate normalized activities (also TOFs) for their catalysts. The current data do not provide evidence for their conclusions. Second, they have to characterize the catalysts after usage to delineate deactivation due to coke deposits and/or Au-NP growth and/or change of SMSI state. Substantially more experimental work is needed to come up with evidence that the SMSI affects – via the Au-NP size – the catalysis.

4. Samples were calcined, most likely in air – please make explicit.
5. Support used is P25 from Evonik. Please provide purity and specific surface area. Have blank experiments been done for the catalysis?
6. Loss versus loose – page 2.
7. What is an ideological thermodynamic equilibrium model?
8. Further checks are needed of the English language.

This paper reports a very interesting phenomenon that, however, needs substantially more data to (dis)prove the model proposed.

Response to reviewers' comments and questions

MS: NCOMMS-20-14885-T

Title: Size-dependent strong metal-support interaction in TiO₂ supported Au nanocatalysts

Author: Xiaorui Du et al

We appreciate all the reviewers for their encouraging comments, valuable questions and constructive suggestions which are very helpful in improving our manuscript. Specific response/answers to each question/suggestion are listed below.

Reviewer #1 (Remarks to the Author):

In this work, the authors present the study on the influence of particle size on metal-support interaction in Au/TiO₂ catalysts. It is claimed that larger Au nanoparticles tend to be encapsulated by TiO₂ overlayers during high-temperature reduction treatment. However, the discussion on the metal-support interaction on the catalytic behavior of various Au/TiO₂ catalysts is not convincing. The control experiments were not well performed. Therefore, I do not think this current manuscript meets the requirements of nature communications and more works are required to reinforce their results and conclusions.

1) Au nanoparticles were prepared by two methods, using different ligands. The calcination temperature is 300°C in air, which may not be enough to completely remove the organic ligands covered on the surface of Au nanoparticles. The authors should make sure the surface of the Au nanoparticles is clean.

Following this comment, the calcination temperature for Au-7nm sample is different from the Au-3nm. It is well known that the geometric structure of metal nanoparticles can evolve during thermal treatment and is related to the gas environment. Indeed, the particle shape of Au nanoparticles of various sizes is different, as indicated by TEM images. So, how the geometric structure may influence the SMSI?

Response:

The reviewer raised a very good question which had actually been taken into account in our original research. To study size effect in nanocatalysts, two strategies are generally adopted: i.e., using same capping agent followed with calcination at different temperatures thus rendering the as-synthesized NPs growing into different size (*Angew. Chem. Int. Ed.* 2017, 56, 2709-2713; *Small Methods* 2018, 2, 1800273), or using different capping agents to obtain as-synthesized NPs with different sizes and then treating at similar temperature to remove the capping agents (*Nature* 2008, 454, 981-983; *J. Phys. Chem. C* 2008, 112(28), 10515-10519; *Nanoscale* 2010, 2, 2798-2804). To ensure that our results are reliable and general, both strategies had been used in our original work, i.e., 3 nm and 7 nm Au NPs were synthesized by same capping agent (PVA) but calcined at different temperatures; on the other hand, 9 nm and 13 nm Au NP was synthesized by using another capping agent oleylamine (OA) and calcined at 350 °C and 400 °C, respectively. It is glad to find that all Au NPs, regardless of the protecting agents or calcination procedures, perfectly follow the same capability order of $13\text{ nm} \geq 9\text{ nm} > 7\text{ nm} > 3\text{ nm}$ in forming SMSI. We believe this is not a coincidence and thus believe neither the types of the protecting agents (which had actually been removed, see discussion below) nor the different calcination procedures have significant effect on the study and the conclusion.

As to the issue of possible residual ligands, we would note that the temperatures used in this work (300 °C for PVA and 350 °C for OA, respectively) are chosen on the basis of previously reported literatures where the temperatures were proved to be effective to remove these ligands. For instance, it has been reported that the PVA can be effectively removed by calcinations in air at 300 °C (*Small Methods* 2018, 2, 1800273; *Angew. Chem. Int. Ed.* 2010, 49, 5771-5775) or even at 250 °C (*J. Am. Chem. Soc.* 2006, 128, 917-924), while calcination at a temperature higher than 350 °C will cause the increase of Au particles size, which lead to the drop of activity. Similarly, OA can be removed by annealing in air at 300 °C (*Nano Res.* 2008, 1, 229-234; *Science*, 2013, 341, 771-773) or even in the range of 165-250 °C (*J. Am. Chem. Soc.* 2016, 138, 8120-8125; *J. Am. Chem. Soc.* 2013, 135, 16833-16836; *J. Power Sources*, 2007, 164, 472-480; *ACS Catal.* 2012, 2, 1358-1362; *J. Am. Chem. Soc.* 2017, 139, 4052-4061; *Science*, 2014, 344, 495-499; *Nano Res.* 2008, 1, 46-55). Based on these previous studies, we believe that the calcination temperatures used in this work, about 50 °C higher

than the previously reported values, are high enough to remove the ligands. To further convince the reviewer, we performed thermogravimetric (TG) and FT-IR examinations of the calcined samples. As shown in **Figure R1a**, the weight loss is negligible (< 0.4%) after 300 °C for both samples, indicating the inexistence of residual ligands; the slight weight loss below 200 °C should be related to desorption of the surface adsorbed water. In addition, the FT-IR spectra (**Figure R1b** and **c**) show that the bands associated with the C-H stretching vibrations (2850 - 2980 cm^{-1}) in OA (*J. Am. Chem. Soc.* 2017, 139, 4052; *ACS Catal.* 2012, 2, 1358) and PVA (*J. Polym. Environ.* 2013, 21, 472; *J. Non-Cryst. Solids*, 2017, 476, 25) disappeared after calcination, reinforcing the successful removal of the ligands from the supported Au NPs.

Figure R1. (a) Thermogravimetric (TG) analysis of the Au-3nm and Au-9nm samples that have been calcined at 300 and 350 °C under air. The samples were heated to 800 °C in air flow (100 mL min^{-1}) with a ramp rate of $10 \text{ }^\circ\text{C/min}$. (b, c) FT-IR spectra of (b) PVA and (c) OA protected Au/TiO₂ and corresponding calcined sample.

For the question how geometric structure of Au influences the SMSI, we don't have an answer so far. It needs an independent study to investigate the effect of geometric structure which is beyond the scope of this work. However, we would note that in this work, the effect comes mainly from size rather than geometry because it has been reported that the equilibrium shape of *fcc* metals is generally truncated octahedron (*Prog. Surf. Sci.* 2005, 80, 92-116). In previously work (*Angew. Chem. Int. Ed.* 2018, 57, 11289-11293), 2.5-4 nm and 2-7 nm of Au NPs obtained by calcination at 400 °C and 450 °C, were identified as truncated octahedral and hemispherical truncated octahedral shape, respectively. We therefore analyzed the representative HRTEM images of Au-3nm and Au-7nm samples in our work, as shown in

Figure R2. It can be seen that the Au NPs had similar shape of truncated octahedron for Au-3nm and Au-7nm, with Au $\{111\}$ and $\{100\}$ facets mainly exposed, consisting with that in the literature (*Angew. Chem. Int. Ed.* 2018, 57, 11289-11293). Thus it is mainly size effect rather than geometry effect.

We have added the new data and corresponding discussions into the revised manuscript.

Figure R2. Representative HRTEM images and corresponding geometric structure analysis for the Au-3nm and Au-7nm samples.

2) In the formulation of the thermodynamic equilibrium for the encapsulation of Au NPs, it is assumed that the Au nanoparticle is a spherical particle. However, they are not, as can be seen in the TEM images. The authors should try to establish a more realistic model and should take into account the Au-TiO₂ interface which is also related to the size of Au NPs.

Response:

When we established the thermodynamic equilibrium model to explicit the inducement of the size effect in SMSI, the shape of Au NPs was simplified to be spherical, which is necessary and feasible in our opinion to facilitate the derivation and make it easier to understand.

In general, when establishing a mathematically model to express physical or chemical concepts and relationships, a certain degree of simplification of the realistic system is

inevitable which is favor to its intelligibility and universality. For instance, in previous studies regarding to the interaction of nanostructured metal overlayers with oxide surfaces, Fu *et al.* (*Surf. Sci. Rep.* 2007, 62, 431-498; *Accounts. Chem. Res.* 2013, 46, 1692-1701) have simplified the oxide supported metal particles system (**Figure R3a**), extracted the mechanical equilibrium as $\gamma_{\text{oxide}} = \gamma_{\text{metal}} \cos\theta + \gamma_{\text{interface}}$, and further expressed the work of adhesion (W_{ad}) as $W_{\text{ad}} = \gamma_{\text{metal}} + \gamma_{\text{oxide}} - \gamma_{\text{interface}}$. This model is efficient to understand the factors that are closely related to metal-oxide interactions, such as adhesion strength, mechanical stability, and fracture behavior of the interfaces (*Surf. Sci. Rep.* 2007, 62, 431-498; *Accounts. Chem. Res.* 2013, 46, 1692-1701). Comparably, in the study of atomistic theory of Ostwald ripening and disintegration of supported metal particles, a simplified model has also been established to explicate the energetics of a supported metal particle (**Figure R3b**), which laid a concrete theoretical foundation for the further in-depth investigation (*J. Am. Chem. Soc.* 2013, 135, 1760-1771).

On the other hand, in our case, the thermodynamic equilibrium model was established to explore the relationship between particle size and the extent of the occurrence of SMSI (the degree of encapsulation), with the size of the most important consideration rather than the morphology of Au NPs (see the response to the previous question). Even if the morphology of Au NPs was set to be non-spherical (such as a truncated-octahedron), and the expression of the radius r in **Figure 4b** may change to a radius of curvature for a certain shape, the fundamental relation in **Eq (1)** will not change and the process of derivation may become complicated but will not be reversed.

In respect of the Au-TiO₂ interface, actually it has already been taken into the thermodynamic equilibrium model: We assumed the encapsulation reaction mainly occurred at the interface of Au/TiO₂, and the interface energy γ_{int} is certainly a key factor in the equilibrium as shown in **Eq (1)**. Obviously, the interface energy is affected by variable r and θ , since it is closely related to the size of Au particles, which was expressed as Eq(S5) in the Supplementary Information document in this case.

Figure R3. (a) Schematic of a metal island on a flat oxide support in thermodynamic equilibrium. γ_{oxide} is the surface free energy of the oxide substrate, γ_{metal} is the surface free energy of the metal overlayer, $\gamma_{\text{interface}}$ is the interface free energy including any metal-oxide interaction, and θ is the contact angle between γ_{metal} and $\gamma_{\text{interface}}$. (*Surf. Sci. Rep. 2007, 62, 431-498*) (b) Schematic of supported metal particle in a spheric segment with the radius of curvature R and the contact angle α between the particle and support. γ_{me} , γ_{ox} , and γ_{int} are the surface energies of the metal particle and support, and the interface energy between metal particle and support, respectively. d is the projected diameter of the metal particle on support. (*J. Am. Chem. Soc. 2013, 135, 1760-1771*)

3) It is claimed that Au-3nm sample reduced at 400 °C by H_2 is only partially encapsulated by the support. However, the TEM is not sufficient to provide such a conclusion since the contrast of the TiO_x overlayers are not very clear in TEM images. The authors should carry out EELS mapping to show the coverage of Au nanoparticles, as complementary results to the CO-IR.

Response:

We truly thank the reviewer for this good suggestion. EELS mapping test of Au-3nm-H400 has been further carried out. As shown in **Figure R4**, both the mapping result (**Figure R4c**) and the correspondingly extracted EELS spectra (**Figure R4e**) of Ti on Au NPs suggest that the Au NPs on Au-3nm-H400 were not completely encapsulated (for example in region 4 the Ti signal is obviously weaker than that in 2 and 3, and more importantly, in region 5 there's no detectable Ti signal). The new data and corresponding analysis have been added into the revised manuscript.

Figure R4 (also Figure 3 in revision). EELS mapping analysis of Au-3nm-H400 sample. (a) Survey image for the EELS mapping test, (b) the EELS spectrum image, (c) the EELS elemental map for Ti, (d) the corresponding HAADF-STEM image, and (e) the extracted EELS spectra (background-subtracted) of the selected positions (red mark) in (b), which were also marked by yellow squares in (d). The yellow line in (a-c) marks the interface of Au NP and TiO₂.

4) The hydrogenation of C=C bonds is not a structure sensitive reaction while the hydrogenation of -NO₂ is a structure sensitive reaction. In **Ref.46**, it is shown that, a selective catalyst for hydrogenation of -NO₂ to -NH₂ is more favorable with metal catalysts with a relatively small exposed surface.

In some works from Somorjai's group, there are also some papers on the discussion of the structure-sensitivity of hydrogenation of olefins.

Response:

The reviewer raised a good question. It is indeed generally accepted that the hydrogenation of low-carbon olefins, especially ethylene, is structure insensitive in many early reports (*J. Catal.* 1969, 14, 23-33; *J. Catal.* 1972, 24, 482-492; *Appl. Catal.* 1987, 32, 145-168) including Somorjai's works (*J. Am. Chem. Soc.* 1984, 106, 2288-2293; *Ind. Eng. Chem. Fundam.* 1986, 25, 63-69, *Catal. Lett.* 1990, 7(1), 169-182), despite the fact that it is actually structure sensitive on catalyst with metal size below 1 nm (*Surf. Sci.* 2016, 652, 7-19; *ACS Catal.* 2019, 9, 11030-11041). It should be noted that the structure insensitivity of this reaction is mainly due to the coverage of metal surface by ethylidyne, rendering the metal playing only a minor role (*Ind. Eng. Chem. Fundam.* 1986, 25, 63-69, see **Figure R5**) which will probably not happen on hydrogenation of aryl olefin (e.g., styrene). Actually the hydrogenation of styrene has been experimentally demonstrated to be structure sensitive in

ref. 46 (*J. Am. Chem. Soc.* 2008, 130, 8748-8753, which has been renumbered as ref. 56 in the revised manuscript) whereas the hydrogenation of nitrobenzene is structure insensitive (see Figure 1 in *J. Am. Chem. Soc.* 2008, 130, 8748-8753, list here as **Figure R6**). Of more significance, to say the least, even if the hydrogenation of styrene is structure insensitive, the selective hydrogenation of 3-nitrostyrene is definitely a structure sensitive reaction due to the different adsorption models of the 3-nitrostyrene on different size of metals. As shown in **Figure R7**, on large metal surface both C=C bond and -NO₂ groups can be absorbed and activated while on small metal surface only -NO₂ group can be preferentially adsorbed (*Chem. Rev.* 2020, 120(2), 683-733), resulting a higher selectivity to -NO₂ hydrogenation product. This is in good agreement with our results that the selectivity to 3-aminostyrene is higher on small Au NPs. Therefore we believe our results are reliable and reasonable. However, to be more accurate and to avoid any possible misleading, we have modified the original claim and added this discussion into the revised manuscript.

Figure R5. Schematic representation of the mechanism for ethylene hydrogenation over Pt and Rh(III) single-crystal surfaces. (*Ind. Eng. Chem. Fundam.* 1986, 25, 63-69)

Figure R6. Effect of Pt loading on the intrinsic reaction rate of Pt/Al₂O₃ catalysts in the hydrogenation

of (○) styrene and (□) nitrobenzene. (*J. Am. Chem. Soc.* 2008, 130, 8748-8753) It shows that with decreasing metal dispersion the specific activity for styrene hydrogenation decreased, while no influence of Pt size on the activity for nitrobenzene hydrogenation can be observed.

Figure R7. Possible adsorption patterns of 3-nitrostyrene on catalysts with different geometric structures. (*Chem. Rev.* 2020, 120(2), 683-733)

5) In Table 1, the author shows the reactivity of Au-3+9nm-H400 and Au-3nm-H400. How about the **Au-9nm-400H**? The higher activity of Au-3+9nm-H400 could be caused by the reduction treatment since the Au-3nm and Au-9nm samples were prepared by calcination in air.

Besides, it seems there is a synergistic effect between Au NPs of 3 nm and 9 nm since the Au-3+9nm-H400 sample shows much higher activity than the Au-3nm and Au-9nm. However, the Au-3nm-H400 sample shows lower activity than the Au-3nm, which could be related to the partial encapsulated of Au nanoparticles, according to **Ref.46**. In my opinion, the SMSI effect is not sufficient to explain such results. More detailed studies should be carried out to verify such effects and give a more sound discussion.

Response:

This is also a very good question. We acknowledge that the reviewers are very insightful to find that there is a synergistic effect between 3 nm and 9 nm of Au NPs when they were co-deposited. We had found this phenomenon as well and thought it is very intriguing and attractive. Ever since, we have been performing a systemic study on this and would like to report the detailed results and corresponding mechanism independently later. As a consequence, we don't think the higher activity of Au-3+9 nm is due to reduction treatment but due to the synergetic effect. This can be evidenced by the different degrees of activity

decrease of Au-3nm (from ~15% to ~12%) and Au-9nm (~15% to ~ 3%) after reduction treatment, which resulted from their different encapsulation levels. However, reduction treatment can indeed improve the selectivity to 3-aminostyrene due to the favorable adsorption of -NO₂ group on electron-rich sites (*Chem. Rev.* 2020, 120(2), 683-733; *Angew. Chem. Int. Ed.* 2020, 59, 11824-11829). This is one of the advantages to utilize the SMSI state for the current catalyst system.

We have added the new data and corresponding discussions into the revised manuscript.

Reviewer #2 (Remarks to the Author):

In this work the influence of Au NP size on the degree of H₂ induced SMSI encapsulation by a TiO₂ support was examined. This is an area of catalysis that has seen interest since the 1970's, and recently has seen significant interest. The SMSI encapsulation state on Au nanoparticles was found to form at a higher temperature for 3 nm Au NPs compared to 9nm and 13 nm AuNPs (600°C vs. 400°C in H₂), suggesting that larger Au NPs are more prone to being encapsulated within the TiO_x layer. On the basis of previously reported size dependent changes to the Au surface energy, the authors developed a thermodynamic model that supports a positive correlation between the degree of the SMSI encapsulation and the Au NPs size. The influence of the size-dependent SMSI formation on the chemoselectivity in hydrogenation of 3-nitrostyrene was also explored.

Overall this paper has interesting measurements and findings. The Au particle size dependent phenomenon may not be completely surprising, but if well analyzed the paper could be published in Nature Comm. However, there are issues throughout the analysis that should be thoroughly addressed prior to publication.

1. It would be useful for the authors to integrate the area of the CO stretch for adsorbed species on Au vs reduction T to show the size dependent decrease in CO coverage clearly. This would be better achieved if the gas phase CO background were subtracted. Also, the units on the IR data should be checked (as shown this is not A.U., this has a specific meaning (Anal. Chem. 2007, 79, 10, 3912-3918)).

Response:

We truly thank the reviewer for this very good suggestion according to which we have integrated the area of CO adsorption peaks on various catalysts by subtracting the contribution of CO gas phase peaks. The relation of normalized CO coverage with reduction temperatures was presented in **Figure R8**. The trends of reduction temperature dependent CO coverage on Au NPs with different sizes are clearly identified. As to the units, our knowledge is that the intensity of IR spectra usually has no units because the data obtained in either

Absorbance or K-M model is a ratio or a logarithm of a ratio of the relative reflectance. We used "a. u." as the abbreviations for "arbitrary units", which has been usually used in IR spectra. However, in order to avoid ambiguity, i. e. the abbreviations for "atomic units" is also "a. u.", it has been deleted in the revised manuscript.

The new data analysis has been added in the revised Supplementary Information, and the manuscript was also modified accordingly.

Figure R8. Normalized size-dependent CO coverage following reduction at different temperatures based on results in **Figure 1**.

2. Correlation between the CO band position and EPR signature is used as evidence of the size effect. First, there is a higher weight loading of Au on the larger particle sample—so this is not a fair comparison. Second, the charge transferred to Au will likely delocalize throughout the particle (it is a metal). To have a more “anionic” Au particle of 9 nm diameter compared to 3 nm diameter, 27x more electrons would need to be transferred due to the particle volumes. Given then 2x higher weight loading in the 9 diam case, more than 50x electrons transferred is needed to make 9 nm Au more anionic than 3 nm Au. This would likely cause bulk reduction of TiO₂? This argument relating EPR to CO IR needs work (see Nature Materials volume 15, pages284–288(2016)).

Response:

The reviewer raised a very good question. We would reminder, however, that the current study is significantly different from the mentioned paper. The mentioned work (*Nat. Mat.*

2016, 15, 284-288) studied the electron transfer under electronic metal-support interaction (EMSI) condition while the current work studies the electron transfer under strong metal-support interaction (SMSI) condition. The concept of EMSI is originated from Campbell's description (*Nat. Chem.*, 2012, 4, 597-598) based on Rodriguez's original research in Pt/CeO₂ catalyst system (*J. Am. Chem. Soc.* 2012, 134, 8968-8974) and has been expanded into other systems later (*Angew. Chem. Int. Ed.* 2014, 53, 3418-3421; *ACS Catal.* 2017, 7, 2339-2345; *Nat. Commun.* 2017, 8, 15802). The distinct difference between EMSI and SMSI is that the EMSI naturally forms in some special catalyst systems without any treatment while the SMSI only occurs upon special treatments, such as high-temperature reduction (the classical SMSI), high-temperature oxidation (*J. Am. Chem. Soc.* 2012, 134, 10251-10258; *J. Am. Chem. Soc.* 2016, 138, 56-59; *Chem. Sci.*, 2018, 9, 6679-6684), or during reaction (*Nat. Chem.* 2017, 9, 120-127; *Nat. Catal.* 2018, 1, 349-355). In this work, we studied the classical SMSI where high-temperature reduction by H₂ is needed, thus the electron transfer between the support and Au is much easier than that in the case of EMSI. In addition, considering that the metal is only about 1 wt% in weigh percent, meaning an even lower percentage in mole ration, we believe the determining factor is how easy the SMSI occurs rather than whether the amount of electron is enough to transfer. In fact, the experimental results (FT-IR spectra of CO adsorption) suggested that larger Au NPs had more negatively charged surface chemical state which is in contrast to the case of EMSI, proving our conjecture. We have added this discussion into the revised manuscript.

3. I was surprised not to see any discussion of recent work from Pd/TiO₂ that suggests a specific structure of TiO_x overlayers on these metals and shows their thermodynamic stability (*Nano Lett.* 2016, 16, 7, 4528-4534). Is the overlayer structure expected to be the same for Au? This should be critical for defining the free energy of the encapsulated state. Also, the model presented is for a rounded Au surface. In actuality is it the higher density of steps and edges for smaller sizes that causes the apparent Au size dependent SMSI behavior?

Response:

We thank the reviewer very much for bring this nice literature and for the good questions. By combining state-of-the-art *in situ* TEM characterization and DFT calculation, the

mentioned literature (*Nano Lett.* 2016, 16, 7, 4528-4534) reported an atomic description of SMSI phenomenon on Pd/TiO₂ system (with 7~8 nm of Pd NP size). One of the main conclusions is that after high temperature reduction, crystalline ordered TiO_x layers formed epitaxially with the Pd(111) surface, and the detailed structure was estimated as one or two atomic layers thick, TiO_{1.5} or Ti₂O₃ respectively. However, it should be noted that the crystalline ordered TiO_x layers only formed under the *in situ* TEM examination process whereas amorphous layer exhibited on Pd NP after *ex situ* reduction under the same condition (*Nano Lett.* 2016, 16, 7, 4528-4534). According to the HRTEM and EELS data in the current work (**Figure 2-3, Supplementary Figure 7**) and our previous report (*Sci. Adv.* 2017, 3, e1700231), the cover layer on Au NPs is also amorphous TiO_x with a few atomic layers thick. Therefore, we believe the cover layer on Au NPs should not be much different from that on Pd NPs, if not exactly same. However, in this work we mainly focused on the experimental demonstration and theoretical explanation of this size-dependent SMSI phenomenon rather than a comprehensive description of this process. Therefore, we acknowledge that an *in situ* TEM study would provide some different, maybe much more valuable information on this novel finding. We believe, however, this can be carried out independently, perhaps by other teams as we are not good at *in situ* TEM study.

As to the question of Au NP shape and the calculation model, it is actually very similar to questions 1 and 2 of reviewer #1. The answers can be found in the responses to those questions. We have performed a detailed structure analysis of Au-3nm and Au-7 nm and found that they actually have same geometry of truncated octahedron structure with Au {111} and {100} facets mainly exposed (**Figure R2**). Thus we believe it is mainly a size dependence rather than shape dependence in our work.

This discussion has been added into the revised manuscript.

4. The reactivity measurements section needs additional discussion. Most importantly, there is a discrepancy between the catalytic conversions measured for Au-3nm and Au-9 nm to that of the Au-3nm+9m mixture after treatment at 400°C, which could be clarified. The catalytic conversions are almost the same for the 40 mg Au-3nm with 0.51% Au and 20 mg Au-9nm with 1.45% Au (entry 1 and 2, Table 1). While the active surface area measurements are

missing, it seems that based on the total Au loading the active site for Au-3 is 1.42 that of the Au-9 nm (a reasonable assumption). However, for the Au-3nm+Au-9nm mixture consisted of 0.25% Au-3 nm (and 1.21% Au-9 nm based on the total 1.46% Au) after the formation of the SMSI state following reduction at 400oC, which substantially covers the active sites on the 1.21% Au-9nm, why did the conversion increase to 28.3%? It would be useful to see the reactivity of the Au-3nm+Au-9nm mixture sample without 400 oC pre-treatment to show it was a simple additive mixture of the behavior of the individual samples. Also, showing the reactivity of the Au-9nm sample after 400 oC reduction would be useful. This and added discussions based on exposed surfaces area of each particle size would be very helpful to clarify the influence of size dependent SMSI on reactivity.

Response:

The reviewer raised a very good question. In our original work we mainly focused on experimentally proving and theoretically explaining the observed size-dependent SMSI phenomenon rather than its catalytic application. Therefore, we used a relatively simple model reaction to try to exemplify the potential application of the novel discovery in catalysis area, just like what we have done in our recent work about different SMSI occurred on Pt single atoms and Pt nanoparticles (*Angew. Chem. Int. Ed.* 2020, 59, 11824-11829), thus didn't go to more details. However, it seems that all the reviewers are very interested in the detailed catalytic performance of the catalysts with or without the occurrence of SMSI as they all raised similar questions. To answer these questions and address the reviewers' concern, we performed more catalytic tests and made more discussions in the revised manuscript.

We would like to stress that, according to the feature of this probe reaction, we focus on the influence of this size-dependent SMSI effect on the reaction selectivity rather than activity. As show in **Figure R7**, on larger metal surface both -NO₂ group and C=C bond can be adsorbed and activated thus a high selectivity to one-group-hydrogenation product is hard. However, on small metal surface, one-group-hydrogenation product will be preferentially obtained (usually -NO₂ group is preferred due to its stronger adsorption, *Nat. Commun.* 2014, 5, 5634). As to the activity, it is quite complex because the activation of H₂ could be the rate determining step which might be different on different size of metal surfaces. Of more importance, in the current work, we have unexpectedly found a synergistic effect of 3 nm + 9

nm Au in terms of activity, which is hard to be clearly explained at present state (see response to question 5 of reviewer #1). Therefore, in the current work we would not focus on the effect of size-dependent SMSI on activity. But from the reaction data we can conclude that the reviewer is correct that the encapsulation can result in deactivation. In addition, by comparing the selectivity of all samples we can conclude that selective encapsulation of 9 nm Au can clearly increase the catalytic selectivity: e.g., without reduction the selectivity of Au-3+9 nm is between those of Au-3 and Au-9 nm, while after reduction the selectivity of Au-3+9 nm-H400 is completely same to that of Au-3nm-H400.

We have added both the reactivity of the mixture sample Au-3+9nm and the Au-9nm-H400 sample, and corresponding discussions into the revised manuscript.

Figure R7. Possible adsorption patterns of 3-nitrostyrene on catalysts with different geometric structures. (*Chem. Rev.* 2020, 120 (2), 683-733)

Reviewer #3 (Remarks to the Author):

Au/TiO₂ catalysts with different Au particle sizes have been studied for the SMSI effect. This effect involves (partial) coverage of the metal surface with TiO_x suboxides to affect the number and performance of active gold metal sites. It is reported that the larger the Au particle size the stronger the SMSI effect, i.e. the lower the reduction temperature at which the SMSI is substantial. This finding in itself is very surprising and if substantiated is sufficient to warrant publication in Nature Communications. TEM, EELS, EPR, CO-FTIR seem convincing in establishing the extent and nature of the SMSI as function of Au-NP size and of the reduction temperature. However, some concerns have to be addressed before publication.

1. As mentioned, it is very surprising that SMSI would be more pronounced for larger Au-NP. This finding is in contrast with conclusion from most recent review – **Nat. Catal. 2 (2019) 955-970** where it is stated that for particles smaller than 4 nm SMSI effects are larger. The authors might want to include this in their introduction.

Response:

We thank the reviewer for bring this nice review paper. This question is actually very similar to question 2 of reviewer #2. We would note that the mentioned work (*Nat. Mater.* 2016, 15, 284-288; *ACS Catal.* 2018, 8, 6203-6215; *J. Catal.* 2018, 367, 194-205) in the review paper actually focused on EMSI. This is significantly different from our work that studies SMSI. The concept of EMSI is originated from Campbell's description (*Nat. Chem.* 2012, 4, 597-598) based on Rodriguez's original research in Pt/CeO₂ catalyst system (*J. Am. Chem. Soc.* 2012, 134, 8968-8974) and has been expanded into other systems later (*Angew. Chem. Int. Ed.* 2014, 53, 3418-3421; *ACS Catal.* 2017, 7, 2339-2345; *Nat. Commun.* 2017, 8, 15802). The distinct difference between EMSI and SMSI is that the EMSI forms naturally in some special catalyst systems without any treatment whereas the SMSI only occurs upon special treatments such as high-temperature reduction (the classical SMSI), high-temperature oxidation (*J. Am. Chem. Soc.* 2012, 134, 10251-10258; *J. Am. Chem. Soc.* 2016, 138, 56-59; *Chem. Sci.*, 2018, 9, 6679-6684), or during reaction (*Nat. Chem.* 2017, 9, 120-127; *Nat. Catal.* 2018, 1, 349-355). Accordingly, the size-dependence in the mentioned work (*Nat. Mater.* 2016,

15, 284-288; *ACS Catal.* 2018, 8, 6203-6215; *J. Catal.* 2018, 367, 194-205) is merely regarding to the electron transfer upon natural contact of metal and support while in our work the size-dependence is related to not only electron transfer but also mass transport under high-temperature reduction. Therefore, it is not very strange there are different size effects in these two studies with different reaction conditions.

We have added this discussion into the revised manuscript.

2. For rationalizing that larger Au-NP are more prone to SMSI it is stated that larger Au-NP have higher surface energies, see Suppl Fig. 9. This statement is contrary to many papers from which I mention only two: Bulletin of the Russian Academy of Sciences: Physics volume 80, pages698–701(2016) and Nanomaterials 2020, 10, 484; doi:10.3390/nano10030484, the latter paper about Au specifically. Smaller particles inevitably have surface atoms with lower coordination numbers at their surfaces which coincides with higher solid-gas surface tensions and not lower ones. So the authors have a serious problem at hand to prove that all other studies except for quoted refs 42-44 that are from a single research group and do not provide enough evidence in my opinion.

Response:

The reviewer raised a very good question. There is almost a consensus that the surface energy may be size-dependent if the particles become small. However, how the surface energy changes with the particle size, i.e. what's the trend of this size dependence remains controversial both theoretically and experimentally, although a positive correlation of surface energy and particle size seems to dominate (*J. Chem. Phys.* 1948, 16, 758-774; *J. Chem. Phys.* 1949, 17, 333-337; *J. Chem. Phys.* 1949, 17, 118-127; *J. Colloid Interface Sci.* 1981, 80, 528-541; *J. Am. Chem. Soc.* 2014, 136, 4508-4514; *Proc. Natl. Acad. Sci. U. S. A.* 2016, 113, 13582-13587; *J. Phys. Chem. B* 2004, 108, 5617-5619; *J. Comput. Theor. Nanosci.* 2011, 8, 2477-2481). Actually, in one of the references you mentioned (*Bull. Russ. Acad. Sci.: Phys.* 2016, 80, 698-701) a similar trend to ours has been reported, despite the fact that they deal with the particle in size < 3 nm which is beyond the size range in our work. Relatively few works (*Nanomaterials* 2020, 10, 484) have suggested an opposite trend.

To address this controversy, Molleman and Hiemstra have recently performed a

comprehensive study on this topic and presented a new understanding in their elegant paper (*Phys. Chem. Chem. Phys.*, 2018, 20, 20575). They showed many subtle things determining the value of surface tension where people usually pay little attention, but the vital one is that an appropriate scaling needs to be done with the help of thermodynamics, i.e., the temperature and the contribution of surface entropy should be taken into account. At very low temperatures, you are totally right that “Smaller particles inevitably have surface atoms with lower coordination numbers at their surfaces which coincides with higher solid-gas surface tensions and not lower ones”. However, at elevated temperatures the contribution of surface entropy cannot be ignored. According to Molleman and Hiemstra’s work (*Phys. Chem. Chem. Phys.*, 2018, 20, 20575) we estimated the size dependence of surface energy and found it has the same trend to our previous calculated one (**Figure R9**).

Figure R9. Size-dependent surface tension at different temperature of Au particles.

3. My second serious concern is that the catalysis studies are not sufficient to draw conclusions. In fact these studies are of a preliminary nature. First, it is essential to report conversion/selectivity versus time curves by variation of the time that the experiment lasts. From that the authors can draw conversion – time curves and via tangent method calculate normalized activities (also TOFs) for their catalysts. The current data do not provide evidence for their conclusions. Second, they have to characterize the catalysts after usage to delineate deactivation due to coke deposits and/or Au-NP growth and/or change of SMSI state. Substantially more experimental work is needed to come up with evidence that the SMSI affects – via the Au-NP size – the catalysis.

Response:

We thank the reviewer very much for the good question and suggestion. As can be seen in response to question 4 of reviewer #2, in this work we used this probe reaction to detect the effect of the size-dependent SMSI on the selectivity rather than activity. As shown in **Figure R7**, on larger metal surface both $-\text{NO}_2$ group and $\text{C}=\text{C}$ bond can be activated thus high selectivity to one-group-hydrogenation product is hard. However, on small metal surface, only one-group-hydrogenation product will be preferentially obtained (usually $-\text{NO}_2$ group due to its stronger adsorption, *Nat. Commun.* 2014, 5, 5634). As to the activity, it is quite complex because the activation of H_2 could be the rate determining step but the activation of H_2 on different surfaces might be different. Of more importance, in the current work, we have unexpectedly found a synergistic effect of 3 nm + 9 nm Au in terms of activity, which is hard to clearly explain at present state (also see response to question 5 of reviewer #1). Therefore, in current work the measurement and comparison of both time-dependent activity and TOF is meaningless. On the other hand, the comparison of the selectivity on all the samples have clearly demonstrated the effect of selective encapsulation of 9 nm Au: e.g., without reduction the selectivity of Au-3+9 nm is between that of Au-3 and Au-9 nm, while after reduction the selectivity of Au-3+9 nm-H400 is totally same to that of Au-3nm-H400.

As to the catalyst stability, we have performed cyclic stability test of the sample. The results have revealed that the sample is stable as in 5 cycles of reaction the activity decrease is minor and selective change is undetectable, **Figure R10**. We have also characterized the used sample by HAADF-STEM and HRTEM. The comparison of HAADF-STEM and HRTEM images of the Au-3+9nm-H400 sample before (**Figure R11**) and after (**Figure R12**) reaction revealed that, the increase of Au particle size was hardly observed. Meanwhile, the situation that large Au particles were encapsulated while the small particles were only partially covered (**Figure R11**) was kept unchanged after the reaction.

All these new data and corresponding discussions have been added into the revised manuscript.

Figure R10. The conversion of 3-nitrostyrene and the selectivity of 3-vinylaniline during five recycling runs.

Figure R11. The HAADF-STEM image (top left) and corresponding Au particle size distribution (top right) of Au-3+9nm-H400 sample; and the representative HRTEM image of the large (bottom left) and small (bottom right) particles in Au-3+9nm-H400.

Figure R12. The HAADF-STEM image (top left) and corresponding Au particle size distribution (top right) of Au-3+9nm-H400 sample after hydrogenation reaction; and the representative HRTEM image of the large (bottom left) and small (bottom right) particles in Au-3+9nm-H400 after hydrogenation reaction.

4. Samples were calcined, most likely in air – please make explicit.

Response:

Yes, before SMSI study all samples were calcined in air at different temperature to remove the ligands (PVA and OA) or improving the sintering of Au NPs (3 nm to 7 nm and 9 nm to 13 nm). Details see the response to question 1 of reviewer #1.

5. Support used is P25 from Evonik. Please provide purity and specific surface area. Have blank experiments been done for the catalysis?

Response:

The P25 (TiO₂) was purchased from Evonik Degussa, with the purity of 99.5%. The specific surface area of P25 was measured by N₂ nitrogen adsorption-desorption isotherms at 77 K using a Micromeritics ASAP-2460 analyzer. Before measurements, the samples were degassed in vacuum at 300 °C for at least 5 h. The specific surface area was calculated by the

BET method, and is $58 \text{ m}^2 \text{ g}^{-1}$.

The blank experiment for the catalysis was carried out, and the results were mended in the revised Supplementary Information documents.

6. Loss versus loose – page 2.

Response:

We thank the reviewer's suggestion. The word "loss" was revised as "lose".

7. What is an ideological thermodynamic equilibrium model?

Response:

To explicit the mechanism of the size-dependent SMSI, we focused on the encapsulation reaction occurred in TiO_2 supported Au NPs, and considered the energy changes of Au NP, TiO_2 support, as well as the Au/ TiO_2 interface during the reaction. The thermodynamic equilibrium of the reaction was then established and expressed as **Eq(1)**. It is inevitable to simplify the realistic system to a mathematically model that is easier to express and understand. Our intention for using the word "ideological" is to point out the simplification and idealization of the established model. After careful consideration, we believe that the word "ideological" is unnecessary. Thus the phrase was revised as "thermodynamic equilibrium model" to avoid ambiguity.

8. Further checks are needed of the English language.

Response:

We thank the reviewer's suggestion. The English language was checked and revised where necessary.

9. This paper reports a very interesting phenomenon that, however, needs substantially more data to (dis)prove the model proposed.

Response:

Thank you very much for your encouraging comments and suggestions. With the new data and discussion we believe the current work is much more convincing.

REVIEWERS' COMMENTS

Reviewer #2 (Remarks to the Author):

The authors did a nice job addressing the referee concerns and the paper is close to publishable.

One remaining comment is that a.u. is still used in Fig 1, while the authors include a scale bar. This axis should simply be "absorbance" which is a defined unit (see ACS Energy Lett. 2019, 4, 8, 2005–2006).

Another remaining comment on the reactivity is the non-additive reactivity of the 3 and 9 nm particles. Although I agree and appreciate that the main focus is on selectivity, the reported conversion seems either to be an error or very surprising. The authors may want to consider addressing this prior to publication.

Reviewer #3 (Remarks to the Author):

My concerns have been largely covered although I still think that it would have been better to report normalized activities (TOF etc) rather than focusing on selectivities only.

Response to reviewers' comments

MS: NCOMMS-20-14885A

Title: Size-dependent strong metal-support interaction in TiO₂ supported Au nanocatalysts

Author: Xiaorui Du et al

Reviewer #2 (Remarks to the Author):

The authors did a nice job addressing the referee concerns and the paper is close to publishable.

One remaining comment is that a.u. is still used in Fig 1, while the authors include a scale bar. This axis should simply be "absorbance" which is a defined unit (see ACS Energy Lett. 2019, 4, 8, 2005–2006).

Another remaining comment on the reactivity is the non-additive reactivity of the 3 and 9 nm particles. Although I agree and appreciate that the main focus is on selectivity, the reported conversion seems either to be an error or very surprising. The authors may want to consider addressing this prior to publication.

Response:

We truly thank the reviewer for the encouraging comments and kind remainder. Indeed the intensity of IR spectra (in Absorbance or K-M model) usually has no units. We have deleted "a. u." in Figure 1 in the revised manuscript.

As to the non-additive reactivity of 3 and 9 nm particles, we fully understand your concerns. Actually we had had the same concerns and have been working on this for a long time. We had performed the catalytic test for many times in the past year and the same phenomenon (the presence of a synergistic effect between 3 and 9 nm) was always obtained although the activity increments are not always exactly the same. Therefore, we believe, at least by far, this should not be an error. We are performing a systematic investigation to try to reveal the generality of this phenomenon (for example for other metal NPs or for other reactions), and more importantly, the mechanism (or underlying reasons) of this phenomenon. We believe this would be a big deal and could not be finished in a short time. Of more importance, we believe this is beyond the scope of the current work which mainly focuses on the phenomenon and mechanism of size-dependent SMSI.

Reviewer #3 (Remarks to the Author):

My concerns have been largely covered although I still think that it would have been better to report normalized activities (TOF etc) rather than focusing on selectivities only.

Response:

Thank you for your suggestion. To address the concern about the TOF, we further performed more catalytic tests. To measure the TOF more accurately, 3-nitrostyrene conversion was controlled below 10% at a shorter reaction time (0.5-4 hours). The TOF was calculated based on the total amount of Au. The corresponding results were listed in Table R1, which have also been added into the revised manuscript and Supplementary Information (Supplementary Table 3).

Table R1. Chemoselective hydrogenation of 3-nitrostyrene using different catalysts

Entry	Catalyst	Reaction time (h)	Conv. (%)	Sel. (%) ^d	TOF (mol _{conv.} h ⁻¹ mol _{Au} ⁻¹)
1	Au-3nm-H400 ^a	1.5	5.2	92.5	7.1
2	Au-9nm-H400 ^b	4	3.5	100	1.3
3	Au-3+9nm-H400 ^c	0.5	4.2	96.3	11.9

Reaction conditions: $T = 110$ °C, $P_{H_2} = 1.0$ MPa; 3 ml reaction mixture: 0.39 mmol of 3-nitrostyrene, toluene as solvent, o-xylene as internal standard.

^a 40 mg of catalyst, with Au loading of 0.51%.

^b 20 mg of catalyst, with Au loading of 1.45%.

^c 20 mg of catalyst, with total Au loading of 1.46%.

^d the selectivity for 3-vinylaniline.